# OpenReview forum: "Self-Alignment with Instruction Backtranslation"
_ICLR.cc/2024/Conference — ICLR 2024 oral_

### Official Review · Reviewer_Fnnk · 2023-10-28

**Soundness:** 4 excellent
**Presentation:** 4 excellent
**Contribution:** 3 good
**Rating:** 8
**Confidence:** 4

**Summary:**

This work investigates scaling the instructing tuning with limited seed data.
The authors suggest an iterative self-training approach to increase training instances from large-scale unlabeled data.
They first train a instruction generation model (backward model) with high-quality seed data to predict instruction for unlabelled data.
Then, the initial instruction following model, which is finetuned on the seed data, is prompted to score the pseudo labeled instances.
Next, a new instruction model is trained on the compound of seed data and selected high-quality pseudo data with system prompt conditioning.
This new improved model can continue the next cycle scoring on the pseudo labeled data, then finetuning the second improved model, again and again.
The pseudo data is not updated in the above iteration.
The authors conducted extensive experiments using LLaMA 7B, 33B, 65B models.
Model performance is evaluated by the win rate of each model against text-davinci-003 from GPT-4 judgements (AlpacaEval).
The generated instructions can increase the task diversity, show better data scaling coefficient than other data sources.
Models finetuned on the selected data achieved best performance among non-distilled models.

**Strengths:**

1.  The paper cleverly utilizes the traditional self-training method to enhance the instruction data and model performance.
2.  The experiments are solid.

**Weaknesses:**

no significant negative issues.

**Questions:**

1. section 3.3, Data quality vs. data quantity. "We find that training on augmented data without self-curation **does not improve** instruction following performance despite scaling up data quantity". I did not find any clear evidence from Figure 2 to support the statement "does not improve." Perhaps adding the win rate of the M0 model could help me better understand?

2. What does the ± in Table 5 mean? Multiple inference of models?

3. I am curious about the performance if we update the backward model using augmented data. It's worth exploring to see how it would perform.

---

> ### Author Response · Authors · 2023-11-21
>
> Thank you for the insightful review and recognition of the impact of the work. The clarification questions are helpful in improving the paper to be more clear:
>
> > 1. section 3.3, Data quality vs. data quantity. "We find that training on augmented data without self-curation **does not improve** instruction following performance despite scaling up data quantity". I did not find any clear evidence from Figure 2 to support the statement "does not improve." Perhaps adding the win rate of the M0 model could help me better understand?
>
> Thanks for the suggestion of adding the win rate of M_0 model, which is 53.57%. By “does not improve” we were referring to the performance of w/o self-curation (the first bar in each group) did not increase as the amount of data is increased, as opposed to the increasing trend with self-curation (the second and third bar in each group).
>
> > 2. What does the ± in Table 5 mean? Multiple inference of models?
>
> It refers to the standard error of win rate, averaged over different prompts where each prompt only has one sample of generation.
>
> > 3. I am curious about the performance if we update the backward model using augmented data. It's worth exploring to see how it would perform.
>
> We did not experiment with using the augmented data to further improve the backward model, i.e. making the augmentation step iterative but it sounds a very interesting idea for future work.

---

### Official Review · Reviewer_XACc · 2023-10-30

**Soundness:** 3 good
**Presentation:** 4 excellent
**Contribution:** 4 excellent
**Rating:** 8
**Confidence:** 4

**Summary:**

The paper aims to build an instruction following language model by fine-tuning. To collect high quality instruction-response pairs automatically, the paper proposes instruction backtranslation, which first uses a seed dataset to fine-tune to generate instructions given a web corpus, and then fine-tunes a stronger model on the filtered instructions. Experiments show the resulting LM outperforms non-distilled LMs on both generation quality and downstream performance. Analysis shows that the self-curation step is critical in selecting high-quality data which leads to further improvement while simply scaling the data size does not.

**Strengths:**

1.	Instruction-following is an important aspect of applying LLMs in practice, while high-quality labeled data is critical to elicit this behavior from LLMs. The proposed method does not rely heavily on human annotation and could scale the data size with the data quality being guaranteed.
2.	The paper conducts extensive experiments with both human and automatic evaluation to demonstrate the effectiveness of the proposed method across downstream tasks and model size. In particular, the analysis verifies that data quality plays an important role in improving performance when scaling up the data size.
3.	The paper is easy to follow and well-organized.

**Weaknesses:**

1.	The paper assumes that the seed model M0 can somehow provide meaningful evaluation for the generated instructions by just following instructions. This might need further investigation. For example, M0 could be just selecting instructions that are similar to the seeds while discarding other instructions which are still useful but may vary in style or format, etc. Also, the work could consider other filtering methods such as using the language modeling probabilities as the scores or using external models such as those trained with NLI.
2.	The paper assumes that a proportion of the unlabeled data should have the corresponding instructions which is not quite intuitive. One limitation is that this might greatly limit the types of instructions that the backtranslation model can generate. I would suggest a further study to understand the types of segments that do have meaningful instructions and the types of instructions that we could collect from the web corpus.

**Questions:**

1.	Could you show more concrete examples of the generated instructions and the corresponding text segments? That would be helpful for users to understand why certain texts should have an underlying instruction.
2.	What are the benefits brought by the proposed method compared to the distilled method?

---

> ### Author Response · Authors · 2023-11-21
>
> Thanks for your thorough review and great suggestions. We will incorporate them in improving the final version of the paper.
>
> > 1. Could you show more concrete examples of the generated instructions and the corresponding text segments? That would be helpful for users to understand why certain texts should have an underlying instruction.
>
> We provided some samples of the segments and the generated instructions in the response to Reviewer HdQC.
>
> > 2. What are the benefits brought by the proposed method compared to the distilled method?
>
> The proposed method does not rely on an external model for data augmentation and curation.
> The synthetic data generated from the proposed method has human-written text in the responses with model-generated instructions while distilled methods have model-generated text as responses.
> Distillation methods have also been shown to be a ``false promise” as is analyzed in [1], where it appears better at following instructions by mimicking the style of the stronger model but still has a large gap on tasks not supported in the distillation data.
>
> [1] Gudibande, Arnav, Eric Wallace, Charlie Snell, Xinyang Geng, Hao Liu, Pieter Abbeel, Sergey Levine, and Dawn Song. "The false promise of imitating proprietary llms." arXiv preprint arXiv:2305.15717 (2023).

---

### Official Review · Reviewer_bTHF · 2023-11-01

**Soundness:** 4 excellent
**Presentation:** 4 excellent
**Contribution:** 3 good
**Rating:** 8
**Confidence:** 5

**Summary:**

This work proposes a scalable and simple method to curate high-quality instruction data for fine-tuning language models.  Specifically, the proposed method includes two stages: self augmentation, i.e., generate prompts for raw documents, and self curation, i.e., select high-quality augmented data iteratively.  After two rounds of data curation, the constructed data is used to finetune a stronger model, which is demonstrated to outperform non-distilled models. Also this paper presents comprehensive analysis and ablation experiments to show the effectiveness of the proposed method.

**Strengths:**

1). An intuitive and effective method to construct high-quality and diverse instruction data. It will significantly reduce the human annotation efforts or potential bias of distilled data from strong LLMs like ChatGPT.

2). The self-curation step provides continuous data quality improvement in terms of fine-tuned models performances and the diversity of augmented data can complement seed instruction data.

3). The paper is well-written with comprehensive and clear analysis / experiments.

**Weaknesses:**

No obvious weakness but it would be better to clarify the choices of unlabeled data for augmentation.

**Questions:**

1). One scaling law question:  will the performance be stable (not increase) with the increased numbers of augmented data (w/ curation)?

2).  In the first paragraph of Section 3.3,  does $A^{(2)}_{5}$ mean the subset that scores more than 4.5?

3). In Table7, what if the results of Humpback 65B with 5-shot demonstrations?

**Details Of Ethics Concerns:**

n.a.

---

> ### Author Response · Authors · 2023-11-21
>
> Thanks for your insightful review and clarification questions. Please find our answers below:
>
> > 1). One scaling law question: will the performance be stable (not increase) with the increased numbers of augmented data (w/ curation)?
>
> That is a great question. The max amount of data we experimented in this paper is 45k examples, which has already demonstrated better data efficiency than other methods (manually annotation, distillation) at that scale. It is definitely an exciting direction to further scale up the augmented data by orders of magnitude or even apply it to pretraining data.
>
> > 2). In the first paragraph of Section 3.3, does $A_5^{(2)}$ mean the subset that scores more than 4.5?
>
> Yes. That’s correct. It was rounded up for concise notation.
>
> > 3). In Table7, what if the results of Humpback 65B with 5-shot demonstrations?
>
> That is an interesting question. The rationale of the comparisons in Table 7 is that few-shot in-context learning is an alternative approach to enable LLM instruction-following capabilities besides finetuning. Therefore, it is more fair to compare the zero-shot performance of finetuned model with few-shot in-context learning from the based model.

---

### Official Review · Reviewer_HdQC · 2023-11-05

**Soundness:** 4 excellent
**Presentation:** 4 excellent
**Contribution:** 4 excellent
**Rating:** 8
**Confidence:** 4

**Summary:**

This paper proposes a method to generate instruction tuning data from unlabelled data by posing the problem as an ‘instruction back translation’ problem ie. given a piece of text, generate potential instructions that can be answered by the text. This model is learnt by finetuning a base LLM with seed instruction data in the reverse direction. The examples thus generated are filtered using the seed model, and iteratively the seed model is improved with the filtered data. Unlike distillation-based approaches, the data is not generated using an external, more powerful model – rather this is self-augmentation that bootstraps a model's capabilities.

**Strengths:**

* The paper proposes a method for generating diverse, high-quality instruction datasets using a baseline LLM that does not require an external, more powerful LLM.
* The instruction-tuned models so created are better than models trained on small, human-curated corpora and competitive with models trained on data distilled from more powerful models.
* Evaluation on a diverse set of benchmarks shows the generalizability of this method.

**Weaknesses:**

While instruction tuning backtranslation is a useful method, it is not clear how it compares with self-instruct. If the same seed dataset had also been used to generate instruction tuning data from the same base LLM, that would provide a  good comparison. The distilled models considered in the paper have been trained on different seed datasets and use more powerful LLMs for distillation. Although I don’t see this as a serious limitation of the paper, this study would have helped shed light on how the two approaches compare.

**Questions:**

* What is the impact of self-curation iterations? Is there a value to performing multiple iterations? Did you also consider inference of the document collection on an improved reverse model from the data extracted (making the augmentation step also iterative)?
* Can you add some samples of extracted instruction dataset instances to the paper?
* How are the segments selected from ClueWeb. Are entire documents chosen, is there some filtering on criteria like length, etc? Are the inputs to the reverse models entire documents or smaller units like paragraphs?
* Table 5: What is the model size used in this study? On which benchmark dataset? Does this observation hold over different model sizes? Which configuration has been used to report results in the rest of the paper?

---

> ### Author Response · Authors · 2023-11-21
>
> > What is the impact of self-curation iterations? Is there a value to performing multiple iterations? Did you also consider inference of the document collection on an improved reverse model from the data extracted (making the augmentation step also iterative)?
>
> Both are excellent questions. We observed improvements in data curation from the seed model $M_0$ to $M_1$. However, from $M_1$ to $M_2$, there was improvement in recall but drop in precision. Therefore, we did not use $M_2$ to conduct a third  iteration of data curation. We did not experiment with making the augmentation step iterative but it sounds a very interesting idea for future work.
>
> > Can you add some samples of extracted instruction dataset instances to the paper?
>
> Below are a couple of examples:
> ```
> Diamond engagement rings gained in popularity during the Art Deco era with the round old European cut diamond being the favourite.
>
> ### Asscher Cut
>
> The Asscher cut is one of the first patented diamond cuts in the world and was invented by Dutch master diamond cutter, Joseph Asscher of the Royal Asscher Diamond Company in 1902.  Classic asscher cut diamonds are cut into squares and resemble emerald cuts, which are rectangular. Asscher cut diamonds are different to a square emerald cut in that they have larger step facets, a higher crown, smaller table and have more brilliance. The corners are cropped to give the shape an octagonal appearance.
>
> ### Baguette Cut
>
> Although the baguette cut was invented sometime prior to the mid-1500s, it only gained popularity in 1912 when Cartier reintroduced the cut to the modern world. Its elongated, table cut, rectangular shape became highly fashionable in the geometric craze of the Art Deco period.
>
> ### Emerald Cut
>
> The emerald diamond cut emerged as one of the first faceted diamond cuts, third in line after the point cut and the table cut. The cut has a dramatic hall of mirrors effect and was standardised in the 1940s.
> \newline
> ```
> *Generated instruction:*
>     List the most popular diamond cuts in the Art Deco era.
> ```
> Inclusive Sports Coaching provides 1:1 Programs for individuals looking to develop their sporting skills, as well as improve their self confidence and opportunities for social and community inclusion.
>
> We recommend an 8 or 12 Session program to identify areas for improvement and sporting skills, conduct drills and physical activities to work towards specific outcomes, while engaging with the client in areas such as listening, memory retention, cognitive processing, social interaction, encouraging conversations, accepting and giving constructive feedback, and other areas as needed.
>
> At the halfway point we produce a status report on progress, and have found parents/carers often share this with OT's, Physios and Teachers as a way to share information on the individual and provide a strong network of support. At the end of the program we produce a final report, with recommendations for ongoing improvement, potential for progress along the person's chosen sport pathway where applicable, etc.
> ```
> *Generated instruction:* I have a business called Inclusive Sports Coaching. We provide 1:1 sport coaching for people with disabilities. I want to have some materials on hand to give to parents when they enquire about our services. What do you recommend I include in these materials?
>
> > How are the segments selected from ClueWeb. Are entire documents chosen, is there some filtering on criteria like length, etc? Are the inputs to the reverse models entire documents or smaller units like paragraphs?
>
> We parse the warc files of ClueWeb in HTML format to extract segments. Each segment is a tree rooted at a header node, including subtrees from lower-level headers. We applied the following filters before sampling segments:
> Length: total length of text between 600 and 3000 characters.
> Duplications: we remove segments with repetitive sentences by computing jaccard similarity of ngrams from pairs of sentences in the segment.
> We remove segments when containing an empty header or the text is all uppercase, header contains navigation text such as “advertisement”, “forum”, “quick link”, “free newsletter”,  etc.
>
> > Table 5: What is the model size used in this study? On which benchmark dataset? Does this observation hold over different model sizes? Which configuration has been used to report results in the rest of the paper?
>
> These ablations were conducted with the 7B model and the win rates were evaluated on the 250 dev prompts sampled from the combined test prompts from multiple sources. We verified the same trend with the 65B model: combined system prompt: $84.34\pm2.31$, only system prompt for OA seed data: $80.15\pm2.51$. Results in the rest of the paper use the combined system prompt in both training and inference, i.e. the configuration corresponding to the first row of Table 5.

---

### Author Response · Authors · 2023-11-21

We thank all reviewers for their thorough reviews and insightful feedback! We are encouraged that they found the proposed approach to be impactful and a timely contribution to the research area. We appreciate that reviewers found our experiments comprehensive and sound. Below we address specific questions which permit more clarification. We will incorporate all suggested improvements in the final version.

---

### Meta-Review · Area_Chair_oBki · 2023-12-10

**Metareview:**

This paper presents a method to curate an instruction tuning dataset by taking a seed instruction tuned model and scraping web data for such augmentation using a "backtranslation"-like approach inspired by machine translation work from the past.  The method is very effective and shows significant gains on standard open benchmarks focused on instruction tuning.

Strengths:  Very clean, effective method, experimental design is very strong, well written paper, strong support from reviewers.  I think the work can have large impact in this thriving area.

I can't see any substantial weakness.

**Justification For Why Not Higher Score:**

Not applicable.

**Justification For Why Not Lower Score:**

I am in favor of papers that present clean solution to a complex problem.  Instruction tuning is a new area that is very fast moving and this provides a pretty straightforward approach to curate high quality data for such tuning.  The reviewers are overwhelmingly supportive and can't find many weaknesses in the paper.  Hence, I think it may deserve an Accept (oral) decision.

---

### Decision · Program_Chairs · 2024-01-16

Accept (oral)